# A Perspective for Ménière’s Disease: In Silico Investigations of Dexamethasone as a Direct Modulator of AQP2

**DOI:** 10.3390/biom12040511

**Published:** 2022-03-28

**Authors:** Robin Mom, Julien Robert-Paganin, Thierry Mom, Christian Chabbert, Stéphane Réty, Daniel Auguin

**Affiliations:** 1Research Group on Vestibular Pathophysiology, CNRS, Unit GDR2074, F-13331 Marseille, France; tmom@chu-clermontferrand.fr (T.M.); christian.chabbert@univ-amu.fr (C.C.); 2Structural Motility, Institut Curie, Paris Université Sciences et Lettres, Sorbonne Université, CNRS UMR144, F-75005 Paris, France; julien.robert-paganin@curie.fr; 3Service d’ORL et de Chirurgie Cervicofaciale, Hôpital Gabriel Montpied, F-63003 Clermont-Ferrand, France; 4Unité Mixte de Recherche (UMR) 1107 Neurodol, INSERM, F-63003 Clermont-Ferrand, France; 5Aix Marseille University-CNRS, Laboratory of Cognitive Neurosciences, Team Pathophysiology and Therapy of Vestibular Disorders, UMR7291, F-13331 Marseille, France; 6Laboratoire de Biologie et Modélisation de la Cellule, ENS de Lyon, University Claude Bernard, CNRS UMR 5239, INSERM U1210, 46 Allée d’Italie Site Jacques Monod, F-69007 Lyon, France; 7Laboratoire de Biologie des Ligneux et des Grandes Cultures, Université d’Orléans, UPRES EA 1207, INRA-USC1328, F-45067 Orléans, France

**Keywords:** human AQP2, dexamethasone, Ménière’s disease, water channel, water permeability modulation, molecular dynamics

## Abstract

Ménière’s disease is a chronic illness characterized by intermittent episodes of vertigo associated with fluctuating sensorineural hearing loss, tinnitus and aural pressure. This pathology strongly correlates with a dilatation of the fluid compartment of the endolymph, so-called hydrops. Dexamethasone is one of the therapeutic approaches recommended when conventional antivertigo treatments have failed. Several mechanisms of actions have been hypothesized for the mode of action of dexamethasone, such as the anti-inflammatory effect or as a regulator of inner ear water homeostasis. However, none of them have been experimentally confirmed so far. Aquaporins (AQPs) are transmembrane water channels and are hence central in the regulation of transcellular water fluxes. In the present study, we investigated the hypothesis that dexamethasone could impact water fluxes in the inner ear by targeting AQP2. We addressed this question through molecular dynamics simulations approaches and managed to demonstrate a direct interaction between AQP2 and dexamethasone and its significant impact on the channel water permeability. Through compartmentalization of sodium and potassium ions, a significant effect of Na+ upon AQP2 water permeability was highlighted as well. The molecular mechanisms involved in dexamethasone binding and in its regulatory action upon AQP2 function are described.

## Highlights:

AQP2 water permeability is modulated by dexamethasone at physiological concentrations.The interaction impacts water fluxes through a direct interaction with the extracellular surface of the aquaporin.Key interactions implicate conserved residues of the ar/R constriction.New insights on corticosteroids mode of actions in Ménière’s disease treatment.The nature of the cations significantly impacts AQP2 water permeability.

## 1. Introduction

Ménière’s disease (MD), first described by Prosper Ménière [1], is a chronic illness characterized by intermittent episodes of vertigo associated with fluctuating sensorineural hearing loss, tinnitus and aural pressure [2].

The main histopathological trait correlating with most MD cases is endolymphatic hydrops (EH), a dilatation of the fluid compartment of the endolymph. It was described by Prosper Ménière in 1861, and was then reported on cadaveric specimen of inner ears from deceased MD patients [3], and is currently visualized in numerous MD patients by means of magnetic resonance imaging of the diseased ear (hydrops found in 84% of MD ears [4]). This phenomenon is now considered to be a marker of pathology [5,6], although its direct involvement in the symptoms of the disease remains to be proven [7,8]. The inducing factors of the EH are still unknown. However, it is believed to result from a massive influx of water from the perilymph, the liquid compartment adjacent to the endolymph [9,10]. Different causes have been put forward to explain this phenomenon, without any being proven.

Aquaporins are transmembrane proteins (Figure 1A) which allow the transfer of water molecules across the biological membranes of most living cells [11,12,13]. Among them, aquaporin 2 (AQP2) has particularly aroused the interest of researchers due to its role in the control of transmembrane water fluxes [14,15,16] and because it is expressed in areas of fluid exchange between the perilymph and endolymph [10,15,17,18,19,20]. A conformational state of the AQP2 tetramer corresponding to open and functional water channels indeed allows, under the effect of the electrochemical gradient, a transfer of water between the two adjacent fluid compartments of the inner ear. This peculiarity makes it a prime target to explain the phenomenon of EH, in the event of a mutation that would affect the structure or the function of AQP2 [21,22].

Different pharmacological approaches have been developed in an attempt to absorb the accumulation of water responsible for EH. The administration of diuretics is part of the MD patient care recommendations [23] to reduce hydrops and alleviate the associated vertigo syndrome. Transtympanic administration of dexamethasone is also one of the therapeutic approaches recommended in MD, when conventional antivertigo treatments have failed [23]. This approach has shown significant benefits in the treatment of MD [24,25,26,27]. Several mechanisms of action have been put forward to explain the effect of dexamethasone in the treatment of MD, such as an anti-inflammatory effect, as in the context of acute peripheral vestibulopathy, or a direct modulation of fluid fluxes at the base of the EH formation [26,28,29,30]. However, none of these hypotheses have so far been theoretically substantiated and verified by experiment.

In the present study, we tested the hypothesis of a direct interaction of dexamethasone with AQP2. Through this interaction, AQP2 water transport activity would be impaired, leading to changes in water fluxes in the inner ear. To test this hypothesis, we addressed the likelihood of this interaction and looked at the underlying molecular mechanisms through molecular dynamics simulations.

## 2. Materials and Methods

### 2.1. Molecular Dynamics Simulations

All simulations were performed with Gromacs (v.2018.1) [31] in a CHARMM36m force field [32]. The systems were built with CHARMM-GUI interface [33]. A first minimization step was followed by 6 equilibration steps, during which, restraints applied on the protein backbone (BB: N CA C O), side chains (SC: side chains heavy atoms) and on lipids (LIPID: polar head heavy atoms) were progressively removed (energies are given in KJ/mol/nm^2^. Phase 1:_BB = 4000.0 _SC = 2000.0 LIPID = 1000.0; Phase 2: _BB = 2000.0 SC = 1000.0 _LIPID = 400.0 -; Phase 3: BB = 1000.0 SC = 500.0 LIPID = 400.0; Phase 4: BB = 500.0 SC = 200.0 LIPID = 200.0; Phase 5: BB = 200.0, SC = 50.0, LIPID = 40.0; Phase 6: BB = 50.0,_SC = 0.0, LIPID = 0.0) before the production phase performed without restraint. Pressure and temperature were kept constant at 1 bar and 310.15 Kelvin, respectively, using Berendsen method during equilibration and Parrinello–Rahman and Nosé–Hoover methods during production. The Lennard-Jones interaction threshold was set at 12 Angströms (Å) and the long-range electrostatic interactions were calculated through the particle mesh Ewald method.

Three experimental setups were carried out.

First, in order to find an interaction site (setup n°1), four molecules of dexamethasone were manually placed in the extracellular vestibules of the tetramer (one dexamethasone per monomer) of human AQP2 (PDB code: 4nef, Figure 1A). The dexamethasone molecules were extracted from the structure of a mouse glucocorticoids receptor (PDB code: 3mne) and positioned by hand in the AQP vestibules in order to mimic the native interactions described by this structure. Then, the tetramer was inserted into POPC bilayer, solvated with 24,641 TIP3 waters and 150 mM of KCl. The system was then simulated during a time course of 60 nanoseconds (ns). The results corresponding to this first experimental setup are displayed in the first Figure of the Section 3.

Then, we rebuilt a tetrameric assembly of AQP2 by duplicating the subunit of the previous system (setup n°2), which interacted with the dexamethasone over the 60 ns of trajectory. The chosen conformation of this subunit corresponds to time = 48.6 ns. This starting conformation was chosen based on two criteria: the high number of hydrogen bonds between the dexamethasone and the protein (three hydrogen bonds) and the presence of the arginine of the aromatic/Arginine (ar/R) constriction within these hydrogen bond contributors (Section 3.1). Two tetramers were built this way: one where we kept the dexamethasone from the previous simulation in interaction with the duplicated subunit; and one where we kept the protein only. Two conditions are obtained this way: one with the chosen subunit duplicated into a new tetrameric assembly without dexamethasone called “control”; and one with the same tetrameric assembly with four dexamethasone molecules interacting with the four extracellular vestibules of the AQP (Section 3.2.1) called “DEX”. Both tetramers were then inserted into POPC bilayer, solvated with 23,071 (system with DEX) and 23,155 (system without DEX) TIP3 waters and 150 mM of KCl. Both systems were then simulated during a time course of 250 ns. The results corresponding to this experimental setup are displayed in Figures within the Section 3 (Section 3.2.1, Section 3.2.2 and Section 3.2.3).

Finally, to estimate the likelihood of the AQP2–dexamethasone interaction further, we built a third system (setup n°3) (Figure 1B): we used the same starting tetrameric assembly as for the second experimental setup condition “control” (i.e., without the dexamethasone fixed inside the extracellular vestibules). The tetramer was then inserted into the POPC bilayer and solvated with 113,916 TIP3 waters. An additional POPC bilayer without AQP was fused to the initial system to compartmentalize it. Then, in the extracellular compartment, 150 mM of NaCl was added, and in the intracellular compartment, 150 mM of KCl was added. To mimic a membrane potential, an additional NaCl ion pair was placed asymmetrically: the Na+ ion was placed in the extracellular compartment and the Cl- ion in the intracellular compartment [34,35]. As a result, a membrane potential of −30 mV was yielded (Figure 1B). Finally, four molecules of dexamethasone were solubilized into the extracellular compartment and four others were solubilized into the intracellular compartment. The system was then simulated for an additional equilibration time course of 5 ns (without restraint) followed by a production phase of 250 ns. The results corresponding to this experimental setup are displayed in Figures within Section 3.3.

### 2.2. Analysis

#### 2.2.1. Water Permeability

To monitor water molecules displacement along the trajectories, the MDAnalysis library was used [36,37]. From these water coordinates, the water count and permeability coefficients (*pf*) were derived. Permeability coefficients were calculated according to the collective coordinate method [38] (see Appendix A for more details). In the present approach, based on the free energy profiles, a correction (described in our previous work) is applied on *pf* to integrate better free energy barriers [34] (see Appendix A for details).

#### 2.2.2. Free Energy Profiles

Water free energy profiles were extrapolated from the logarithm function of the water counts inside the pore with the *z*-axis as a reaction coordinate [39,40]. The pore is divided along the reaction coordinate (*z*-axis) in slices of 0.5 Angströms (Å). The average density of water molecules in each slice is then computed over the 250 ns of simulation and the Gibbs free energy *G(z)* is obtained as follows:$$G\left(z\right)=-KTln\frac{\rho \left(z\right)}{\rho_{bulk}}$$
where *K*, *ρ_bulk_* and *T* represent the Boltzmann constant, the bulk density and the absolute temperature, respectively.

#### 2.2.3. Binding Free Energy and Dissociation Constant

The binding affinity of dexamethasone to AQP2 was evaluated directly from the structure, extracted from the molecular dynamic trajectories every nanosecond, with PRODIGY-LIG program [41]. PRODIGY-LIG evaluates the contacts between ligand and protein and computes a free binding energy from a reliable empirical equation [42].

Dissociation constant (*K_D_*) values were obtained from the binding free energies as follows:$$K_{D}=exp\left(\frac{-\Delta G_{S}}{RT}\right)$$
with Δ*G_S_*, *R* and *T* indicating binding free energy, the ideal gas constant, and the temperature, respectively.

#### 2.2.4. Other Properties

Membrane potential, hydrogen bonds, distances, radial distribution functions (rdf) and dipole moments are computed with Gromacs tools (version 2020.6). Pore profiles are computed with HOLE software [43].

#### 2.2.5. Statistical Analysis

All statistical analyses were performed using R programming language. Before any statistical test was performed, normality and homoscedasticity of the variables were controlled to choose between parametric or non-parametric tests. When two variables were compared, Student’s t-test or Mann–Whitney test was used. When more than two variables were compared, Tukey’s post hoc test after one-way analysis of variance or Bonferroni post hoc correction after Wilcoxon test was used.

For experimental setup n°2, the 250 ns trajectories were divided into 10 ns subtrajectories and the analyses were performed for each monomer, hence yielding 100 repetitions per condition for Figures 3 and 5, and 25 repetitions per monomer for Figure 4. For the contingency table shown in Figure 4D, the functional state of a channel was established when allowing 5 or more water molecules to cross the whole 30 Å long transmembrane section from 1 compartment to another in 10 ns. This threshold was defined based on the results of permeability comparison between the condition “control” and “DEX” over the 250 ns of trajectory and displayed in Figure 3. The value of 5 water molecules was elected as a functional state threshold because it discriminates both the means and medians of the two conditions.

For experimental setup n°3, the 250 ns trajectory was divided into 10 ns subtrajectories and the monomers were compared between each other, yielding 25 repetitions per condition for Figures 6 and 7B. In Figure 7D, 40 ns or 50 ns slices of trajectory only were used for analysis, hence yielding 4–5 repetitions per condition. In Figure 8, the system is divided in 10 Å long slices according to the *z*-axis. To estimate the correlation between the dipole moment of the tetramer and the dipole moment of the dexamethasone, the angle they form with the *z*-axis was used. The *z*-axis was chosen as the reaction coordinate because of the native stable orientation of AQP2 tetramer dipole moment, almost parallel to the *z*-axis (Figure 8C). For each slice, the frames in which the centre of geometry of the dexamethasone considered (residue number 228 for extracellular compartment and residue number 232 for intracellular compartment) was contained, the slices were used to extract the dipole moment of the dexamethasone and the corresponding dipole moment of AQP2 tetramer. The Spearman’s rank rho coefficient was then used to estimate the correlation between angle values for each slice.

## 3. Results

### 3.1. AQP2–DEX Interaction Site

First, to find a putative interaction site for dexamethasone in AQP2, we used the X-ray structure of the mouse agonist glucocorticoids receptor interacting with dexamethasone (PDB code: 3mne [44]) as a reference (Figure 2A). From this structure, we can notice the protein residues of the interaction site complement the amphipathic nature of the dexamethasone very well. Indeed, we found both polar and charged residues, such as asparagine, glutamine or arginine, and hydrophobic ones, such as isoleucine or tryptophane (Figure 2A). Tryptophane can also act as a hydrogen bond donor and is hence able to accommodate both polar and hydrophobic parts of the dexamethasone. When we compared this interaction site with the extracellular vestibules of AQP2, we found the above-mentioned features were conserved. On this part of the AQP, an arginine, an asparagine, a glutamine and a tryptophane were found (Figure 2B). Plus, the intrinsic amphipathic nature of the conducting pore [45,46] characterized by the opposition between polar water molecules interaction sites and hydrophobic slides leads to very similar features. This amphipathic nature of the AQP-conducting pore allows very stringent and selective conduction properties [39] but could also accommodate other amphipathic molecules, such as dexamethasone. In addition to this similar nature, with the native interaction site of dexamethasone, the extracellular vestibules of AQPs also present a very conserved residue among the family: the arginine of the aromatic/arginine (ar/R) constriction [11]. This ar/R constriction is named after this arginine and is well known to be determinant in the function of the protein, for example, it is called the selectivity filter of the pore. This constriction corresponds to the narrowest part of the channel and its role in the selectivity of the channel has been confirmed by both theoretical studies [47,48] and mutation experiments [49]. Indeed, the composition of the constriction changes both the size and the hydrophobic nature of the pore which are understood as the two criteria on which relies the selectivity of the AQP [39]. More recently, several studies further confirmed the critical role of the arginine of the ar/R constriction on the transport properties of the AQP by highlighting significant impact of the position of its side chain in the pore on the water permeability [34,35]. Thereafter, because of the similarities between the extracellular vestibules of AQP2 and the interaction site of the dexamethasone and because of the presence of very conserved residues known for their role in transport properties of AQPs, we decided to investigate the ability of these vestibules to fix dexamethasone. In order to do so, we manually placed one dexamethasone molecule on each four extracellular vestibules of the X-ray structure of human AQP2 (PDB code: 4nef [50]). We mimicked the interactions observed on the X-ray structure of the mouse agonist glucocorticoids receptor: the oxygen of the carbonyl group is positioned close to the arginine, the hydroxyl groups close to the polar asparagine and glutamine while the hydrophobic side of the dexamethasone faces the hydrophobic part of the channel ending at the extracellular surface with the tryptophan (Figure 2A,B).

Then, we simulated the system for 60 ns and observed that the interaction was stable over the whole course of the trajectory, at least for one of the subunits. This is illustrated by the number of hydrogen bonds established between the protein and the dexamethasone molecule positioned in the extracellular vestibule of this subunit (chain D) (Figure 2E). All of the residues interacting with the dexamethasone through hydrogen bonding over the 60 ns of simulation are represented in Figure 2F. To obtain a first impression of the impact of such an interaction on the activity of the channel, we counted the number of water molecules crossing the AQP over the whole transmembrane section (30 Å). The results are displayed in Figure 2D and clearly indicate a non-functional conducting pore for the protomer interacting with dexamethasone over the whole trajectory (chain D). We had the hypothesis that, because this interaction includes hydrogen bonding with the arginine of the ar/R constriction (Figure 2F), it would have a significant impact on both the criteria impacting the selectivity of the channel (i.e., size and hydrophobicity), and hence on the permeability of the channel. On top of that, the steric hindrance of the dexamethasone itself inside the pore can probably play a significant role as well. This hydrogen bond between the dexamethasone and the ar/R constriction arginine is visualized in Figure 2C. The interruption of the water molecules continuum was also observed. To further estimate the impact of this putative interaction on the water transport ability of AQP2, a new tetramer was built. We took a conformation of chain D (which is the protomer presenting the most stable interaction) when the ar/R constriction arginine was involved in a hydrogen bond with the dexamethasone, when the number of hydrogen bonds was the highest (three simultaneous bonds) and corresponding to the end of the trajectory (at time = 48.6 ns; in order to start from a well-relaxed conformation). We then duplicated the protomer in order to reconstitute a tetrameric assembly that we used to further characterize this interaction and its impact on water fluxes through AQP2.

### 3.2. Impact on the Transport Ability of the Channel

#### 3.2.1. Dexamethasone Has a Significant Impact on AQP2 Water Permeability

To estimate the impact of the dexamethasone interaction with AQP2 on water transport, we first compared the whole tetramers with and without dexamethasone together over a 250 ns simulation time course (Figure 3). The osmotic permeability coefficients (*pf*) obtained from the simulations (mean *pf* of 2.13 × 10^−14^ cm^3^.s^−1^ derived from the 250 ns of simulation without dexamethasone) were very close to the experimental values (3.3 × 10^−14^ cm^3^.s^−1^) issued from the heterologous expression systems (constructs expressed in *Xenopus* oocytes) [14]. Moreover, the difference appeared very significant between the two conditions (Figure 3A). In a previous work [34], we highlighted a bias that could be associated with the collective diffusion model used to estimate *pf* from molecular dynamics simulations [38]. Hence, to strengthen the characterization of dexamethasone impact on water transport, we computed two other permeability estimators as well (Figure 3A). The number of water molecules crossing a 5 Å long section of the channel containing the ar/R constriction constitute a more straightforward approach [34] and discriminated the two conditions even more significantly (Figure 3A). To be more stringent, we also used the number of water molecules crossing the whole conducting pore spanning the entire transmembrane domain over a 30 Å long section and found the same very significant difference between the two conditions (Figure 3A). These first results point toward a significant impact of dexamethasone interaction with AQP2 extracellular vestibule on its water transport activity. However, from the cumulative number of water molecules crossing the pore over the 250 ns trajectory, it clearly appeared that dexamethasone did not impact the four monomers equally (Figure 3B).

#### 3.2.2. Dexamethasone Impacts AQP2 Water Permeability through Its Interaction with the Arginine of the ar/R Constriction

To better understand this heterogeneity of response phenotype between the four monomers, we focused our study on the way dexamethasone impacts water permeability in the “DEX” condition (Figure 4). We first hypothesized that a high number of hydrogen bonds between the dexamethasone and AQP2 would stabilize the interaction and hence correlate with smaller water permeability. However, we observed the contrary, with a high number of hydrogen bonds between the dexamethasone and AQP2 correlating with high water permeability values (Figure 4A). We also hypothesized that an interaction with the arginine of the ar/R constriction would impact the permeability of the channel as it would modify both its size and hydrophobicity. Indeed, we observed a very strong correlation between the smallest distance between this arginine of the ar/R constriction and the dexamethasone and the permeability of the channel (Figure 4B): the smaller the distance, the smaller the permeability. On top of that, the comparison of pore diameters between the two functional protomers and the two non-functional protomers (chain A and D versus chain B and C) yielded a significant difference at the narrowest part of the pore, i.e., at the ar/R constriction region (Figure 4C). Finally, to establish whether the interaction between the arginine of the ar/R constriction and the dexamethasone through hydrogen bonds formation was correlated with the functional state of the pore or not, a Chi2 test was performed. It strongly pointed toward a statistically significant association between the two variables with a *p* value of 4.95 × 10^−11^ (Figure 4D). A pore was considered functional based on the precedent set of data (Figure 3, see Section 2 for more information). This association is represented graphically in Figure 4E,F.

Finally, we concluded that dexamethasone significantly inhibits water transport through AQP2 through its interaction with the arginine of the ar/R constriction (Figure 4D–F). This interaction corresponds to a conformational state, where the dexamethasone was stabilized close to the ar/R constriction region (Figure 4B), inside the extracellular vestibule, through a small number of hydrogen bonds with AQP2 (Figure 4A). This interaction directly modified the size of the pore (Figure 4C). Because the direct interaction (Figure 4D,F) changed the position of the arginine side chain inside the pore and the radiation of the positive charge of the guanidinium, the hydrophobicity of the pore was impacted as well. Hence, dexamethasone had a significant impact on water permeability of AQP2 through its interaction with the arginine of the ar/R constriction, which, in turn, changed both the crucial criteria behind pore water permeability (i.e., size and hydrophobicity [39]).

#### 3.2.3. Detailed Impact of Dexamethasone Interaction with AQP2 on the Water Permeability as an Illustration of pf Adjustment with the Dk Constant

In a previous work, we highlighted a bias that could arise from the *pf* calculation with the collective diffusion method [34]. Due to the thermal agitation of water molecules inside the pore, an overestimation of water permeability could be derived from closed channels. To tackle this issue, we proposed a correction constant, the *Dk* constant. It allows for a higher integration of free energy barriers into *pf*. Here, we confronted this approach with a new set of data. Figure 5 displays the free energy profiles and water permeabilities of each four protomer without (“control” condition) and with (“DEX” condition) dexamethasone. First of all, by simply counting the water molecules crossing the whole transmembrane section, we can confirm the previously observed pattern: chain A and D did not stabilize dexamethasone in the extracellular vestibule (see previous section) and were in an open functional state in both conditions, while chain B and C, through their interaction with dexamethasone (see previous section) were maintained in a closed non-functional state in the “DEX” condition.

This pattern was also affirmed by the free energy profiles with the apparition of free energy barriers before the ar/R constriction region for chain B and C. A high free energy barrier located at the ar/R constriction was also observed in the control condition of chain D. This could be explained by the propensity of the arginine of the constriction to switch between the up and down conformational states [34,35]. This feature has been observed in several AQPs and seems conserved among the family members [34]. AQP2 especially was recently highlighted as particularly concerned by this phenomenon through molecular dynamics study [40]. However, when *pf* was used to compare the two conditions, significant difference existed only for chain C. We applied the *Dk* constant correction on *pf* (see Appendix A and Appendix A for details) and restored the response phenotypes observed through water counts: a significant difference between “control” and “DEX” conditions can be observed for chain B and C. These results add evidence in favour of the *Dk* correction constant that we introduced in our previous work [34].

### 3.3. Biological Relevance of the Dexamethasone–AQP2 Interaction

#### 3.3.1. Impact of Cation Nature on the Permeability of AQP2

To test whether the interaction between dexamethasone and AQP2 was likely to happen spontaneously in vivo, we built another more realistic atomic system (see Section 2). This new system takes into account the asymmetric partitioning of ions sodium and potassium between the plasma membrane necessary for the establishment of membrane potential. Hence, we placed 150 mM of NaCl in the extracellular compartment and 150 mM of KCl in the intracellular compartment (see Section 2, Figure 1).

We first estimated the impact of cation nature on the dexamethasone and the AQP. The results are presented in Figure 6. Radial distribution functions of sodium, potassium and chloride ions with the dexamethasone molecules as reference showed a very quick convergence toward the equilibrium value of 1, which corresponds to a ideal gas state distribution (Figure 6A). To be sure that the dexamethasone molecules were not in interaction with the protein or the membrane and avoid a potential bias, we used the 5 ns of additional equilibration, when the dexamethasone molecules were in solution. In contrast, a clear impact of the nature of the cation appeared when the chosen reference corresponded to the exposed carboxylates of the extracellular or intracellular surface of AQP2 (Figure 6B). An over-accumulation of cations within a 5 Å range of the protein carboxylates is specific to the sodium but not to the potassium ions (Figure 6B). To explain this phenomenon, we hypothesized that the smallest Van der Waals radius of sodium and its higher electronegativity makes it “stickier” toward negatively charged carboxyl groups.

Among the four chains of the AQP2 tetramer, two interacted with the dexamethasone (chain B and C) and two others did not (chain A and D). We started by characterizing the monomers not interacting with dexamethasone (Figure 6F): chain A water permeability is significantly smaller than chain D and this difference correlates with the size of the ar/R constriction. In a previous work, we demonstrated that the electrostatic environment of the ar/R constriction, at the extracellular surface of the AQP, could impact water permeability by orientating the position of the arginine side chain inside the pore [34]. Hence, we hypothesized that the fixation of sodium ions on the carboxylates of the extracellular surface could induce this conformational change of arginine side chain, reflected by the change in size of the constriction. Therefore, we looked at the radial distribution functions (rdf) of sodium with each extracellular carboxylate as a reference separately. We observed a higher over-accumulation of sodium, around 3 out of 5 carboxylate (aspartate 111, aspartate 199 and aspartate 200), for chain A compared with chain D. We observed the opposite pattern for glutamate 106 and an equal over accumulation between the two chains for aspartate 115. These observations comfort our hypothesis as a higher number of negative charges located at the periphery of the monomer are attenuated in chain A when compared with chain D and correlates with the difference in water permeability. However, we also observed the complementarity of charged residues positioned on the extracellular loops C and E (so-called C-loop and E-loop, respectively) (Figure 1A and Figure 6C). The C-loop is a mobile loop which dives into the extracellular vestibule of AQPs (Figure 1A) and interacts through hydrogen bonds with the arginine of the ar/R constriction implicating alanine 117 and asparagine 119 (Figure 6E). We formulated a complementary hypothesis, stipulating that changes in the position of the C-loop, mediated by salt bridges formation with E-loop residues, could modulate the ar/R arginine side chain position through the hydrogen bond network formed between C-loop and the arginine. Based on the residue composition of AQP2, two salt bridges can be formed to stabilize C-loop position: between glutamate 106 and arginine 113 and between aspartate 115 and lysine 197 (Figure 6C). There is a higher over accumulation of sodium at the vicinity of glutamate 106 of chain D, which indeed correlates with a significantly higher E106-R113 distance. Aspartate 115 interacts equally with sodium between chain A and chain D. However, it is not the case of aspartate 199, which competes for the interaction with lysine 197 as well. Hence, chain A aspartate 199, by interacting more with sodium than its counterpart in chain D, would compete less for the salt bridge formation with lysine 197. This in turn would be in favour of salt bridge formation between lysine 197 and aspartate 115 in chain A, reflected by the smallest distance between these two residues. Even though the C-loop stays within a distance allowing for salt bridge formation in both chains (<5 Å), these significant differences still reflect a probable conformational change. Finally, to estimate the impact of such changes on the position and orientation of the C-loop atoms interacting with the arginine of the ar/R constriction, the angle formed between alanine 117 C-O and asparagine 119 CB-CG with the *z*-axis are computed (Figure 6E,F). A significant difference is observed for alanine C-O orientation. This indicates a torsion of C-loop that could have led to the destabilization of hydrogen bond between alanine 117 and arginine 187. Based on all these preliminary results, we can conclude that the sodium ions, through their interaction with AQP2 extracellular surface carboxylates, change the charges repartition which in turn impacts the water permeability of the channel through the modulation of arginine 187 side chain position inside the pore. The data presented here are not complete enough to estimate correctly whether the electrostatics surface potential modification, the conformational changes of C-loop, or both criteria, can explain this significant impact of sodium interaction on water permeability of the monomer.

#### 3.3.2. Interaction of AQP2 with Dexamethasone without Docking

To evaluate better the likelihood of the dexamethasone–AQP2 interaction, in this third more realistic setup, the four dexamethasone molecules were placed in solution in both cellular compartments. The dexamethasone interacted very quickly with the AQP2 extracellular surface, 11 ns after the beginning of the simulation (see Appendix A), and converged toward the predicted interaction site in about 40 ns, characterized by the formation of hydrogen bond with the arginine of the ar/R constriction (see previous sections, Figure 7C). We previously showed that the formation of such a hydrogen bond was a determinant factor in significantly impacting water permeability (see previous section). Here, we observed this phenomenon on two monomers: chain B and chain C (Figure 7C). Two different conformations correspond to this state and differ between the two chains: one implicates the formation of the hydrogen bond with the arginine and the carbonyl oxygen of the dexamethasone (Figure 7G); the other corresponds to the dexamethasone orientated in an opposite manner, presenting its hydroxyl groups toward the guanidinium group of the arginine (Figure 7G). The first conformation was observed inside the extracellular vestibule of chain C and inhibits water flux better than the second one, found inside the vestibule of chain B (Figure 7D). It also correlates with a more negative binding free energy (Figure 7F).

Even though these conformations impaired the water flux the most significantly (see previous section and Figure 7A,C,D), other ones also stabilized the dexamethasone molecule inside the extracellular vestibule. Indeed, the dexamethasone 228 interacted with AQP2 inside the chain C extracellular vestibule from time = 25 ns to the end of the trajectory (Figure 7E). During this 225 ns uninterrupted interaction, the dexamethasone switched between conformations (Figure 7G). Even though it did not stabilize the whole time in direct interaction with the arginine of the ar/R constriction, overall, it significantly impaired the water permeability of chain C (Figure 7A,B). The binding free energy corresponding to this interaction with chain C ranges from −5.21 kcal.mol^−1^ to −10.12 kcal.mol^−1^ with an average of −9.22 kcal.mol^−1^. It corresponds to dissociation constant values (K_D_) ranging from 212,004.7 nM to 73.15 nM with an average of 317.7 nM. This last set of data clearly demonstrates the spontaneous nature of the interaction and confirms its significant impact on the water permeability of AQP2.

#### 3.3.3. Proposed Molecular Mechanism for Specific Interaction between AQP2 Extracellular Surface and Dexamethasone

During the 250 ns simulation, the four dexamethasone of the extracellular compartment spontaneously established stable interactions with the AQP2 surface (see Appendix A). In contrast, in the intracellular compartment, none of them interacted more than 2 ns with the protein surface. One of them only was quickly stabilized inside the lipidic bilayer and interacted with the hydrophobic transmembrane region of AQP2 for approximately 10 ns (see Appendix A and Appendix A). We formulated the hypothesis that dipole moment orientation of the molecules coupled with charges attractions could explain this difference. On the upper part of Figure 8A,B are displayed the correlations between the dipole moments of AQP2 tetramer and the dipole moments of the dexamethasone molecule which interacted with the chain C extracellular vestibule during 225 ns (Figure 7E). Additionally, on the bottom part of Figure 8A,B are displayed the correlations between the dipole moments of AQP2 tetramer and the dipole moments of one of the three intracellular dexamethasone, not stabilized inside the lipidic bilayer.

In order to do so, the dipole moments orientation was compressed inside a single variable, chosen as the angle formed between the dipole moment and the *z*-axis (see Section 2). The correlation between these angle values computed for AQP2 and the dexamethasone molecule was calculated according to the position of the dexamethasone molecule inside the simulated box. For the extracellular compartment, starting from the tetramer surface, we can observe a very significant correlation between the dipole moments for the first 20 Å, followed by higher *p* values, eventually reaching the point of non-significant correlations in the upper part of the compartment (slices 250–260 and 260–270) before finally going back to a significant association in the last slice (270–280). These observations could be explained by the influence of the dipole moment of the tetramer (1198.98 Debye) upon the smaller dipole (2.93 ± 1.08 Debye) moment of the dexamethasone, correlated with the charge interactions of the molecule with the protein and the membrane. Indeed, in the first 10 Å, the dexamethasone interacted directly inside the vestibule of chain C and was hence stabilized in a preferential orientation. In the next slice however, the very low *p* value could be the result of a synchronization effect of the small dipole moment of the dexamethasone with the three orders of magnitude higher dipole moment of the tetramer. Then, going further from the protein and higher in the simulated system, this effect would be attenuated by the distance. Finally, the last slice small *p* value could be the result of a stabilizing interaction of the dexamethasone with the polar head of the POPC. If the dipole moment of AQP2 orients the dexamethasone molecule, then its negative pole would mostly point toward the upper part of the simulated system, while its positive pole would be, on the contrary, mainly orientated toward the bottom part of the system (Figure 8C). This, correlated with the surface charges of the POPC and AQP2, could explain the opposite behaviour of the dexamethasone between the two cellular compartments: in the extracellular compartment, this phenomenon would orient the negative pole of the dexamethasone toward the POPC polar heads, displaying their positively charged choline groups, and its positive pole toward the negatively charged extracellular surface of AQP2 (Figure 8D). In the intracellular compartment, the exact opposite situation could be expected, as the intracellular surface of AQP2 is still mostly negatively charged. This charge repulsion could explain the fact that the correlation was not significant at the vicinity of the membrane, and that there is not enough data to test the correlation near the AQP2 surface (Figure 8A,B and Appendix A). Finally, on the extracellular side, the neutral centre of the tetramer probably helps to accommodate the hydrophobic side of the dexamethasone molecules before they reach out one of the monomers vestibules (Figure 8D and Appendix A).

## 4. Conclusions and Discussions

In the present study, we have demonstrated the relevance of the AQP2–dexamethasone interaction. The mean binding free energy corresponds to a K_D_ of 317.7 nM, falling in the range of specific interactions [52]. In the last experimental setup, which mimicked a cellular context the best, the dexamethasone placed in solution spontaneously interacted with AQP2, in the short time lapse of 11 ns (Figure 7). Moreover, two dexamethasone molecules converged toward the predicted interaction site without external forces being applied on the system. One of them remained docked inside the extracellular vestibule for 225 ns out of the 250 ns simulated (Figure 7). This spontaneous interaction was specific to the extracellular compartment, while in the intracellular compartment, no interaction was observed (see Appendix A). To explain this asymmetric behaviour of the dexamethasone relative to its position in the system, we formulated a hypothesis implicating the influence of AQP2 dipole moment on the orientation of dexamethasone coupled with charges attraction or repulsion with the protein and the lipids surface (Figure 8). However, more investigations are needed to conclude in favour of this hypothesis. The new interaction site was initially investigated on the basis of the similar features shared with the mouse glucocorticoids receptor X-ray structure (PDB code: 3mne) [44]. The interaction site was located on the extracellular surface of AQP2, inside the vestibule of each monomer (Figure 2). The tetrameric assembly hence displayed four interaction sites for dexamethasone. To estimate the impact of this interaction on the channel water permeability, three different methods were used (*pf*, number of water molecules crossing the ar/R constriction and number of water molecules crossing the whole transmembrane section: Figure 3) in combination with three different atomic systems (initial docking of dexamethasone on the crystallographic structure, reconstructed tetrameric assembly and the same tetramer contextualized inside a compartmentalized system), which all converged toward the same conclusion: there is a significant reduction in water permeability of AQP2 triggered by its interaction with dexamethasone. The dexamethasone reduces water fluxes the most when establishing hydrogen bonds with the arginine of the ar/R constriction (Figure 4). This conformation corresponds to the smallest binding free energy (−10.12 kcal.mol^−1^) and the smallest K_D_ (73.15 nM). Even though the interaction between dexamethasone and the arginine of the ar/R constriction is associated with dexamethasone inserted the deepest inside the AQP vestibule, it occurred spontaneously in about 40 ns (Figure 7). When the dexamethasone is initially placed in this position (Figure 4), we observed the interaction with the arginine maintained stable for 200 ns in two monomers. For the two others, the interaction was observed several times, pointing toward a recurrent phenomenon rather than a rare event. Because the ar/R constriction corresponds to the smallest part of the channel and to an interacting site with water molecules [39,49], the interaction between the arginine 187 of the ar/R constriction and the dexamethasone impacts significantly both the size and the hydrophobicity of the channel (Figure 4). Even though we observed a significant reduction in the pore diameter at the ar/R constriction by dexamethasone interaction, the difference in size was really small (~0.1 Å, Figure 4). Moreover, in the last experimental setup, no significant ar/R constriction size reduction was observed in correlation with dexamethasone fixation (Appendix A). However, the hydrogen bond formed between arginine 187 side chain and the dexamethasone attenuated the positive charge of the guanidinium group, usually radiating inside the pore. This change in electrostatic profile can be estimated through the differences in pore diameter (Figure 4) and free energy (Figure 5) profiles in other regions than the ar/R constriction. From the free energy profiles, a correction (*Dk* constant) was applied to *pf* values, allowing significant differences to emerge between “control” condition and “DEX” condition (Figure 5). While the *Dk* constant was originally designed to integrate better the ar/R constriction free energy barrier into *pf* [34], in the present study, the highest free energy barrier used for integration corresponded to the position between NPA motifs and ar/R constriction (Figure 5 and Appendix A). Taken together, these results suggest that the main contribution of the dexamethasone–AQP2 interaction on water permeability reduction is of an electrostatic nature, and that the energetic constriction does not correlate with the steric constriction of the channel. We also observed further evidence in favour of the regulatory role of charge distribution in proteins through the impact of sodium on AQP2 water permeability (Figure 6). By quenching carboxylates of the extracellular surface located on the periphery of the tetramer, sodium ions induced a net conformational change of the arginine of the ar/R constriction conformation characteristic of a closed channel (Figure 6). This conformation of the arginine, designated as “down state”, was already observed in the crystal structures and molecular dynamics studies, and was linked to the electrostatic environment changes of the protein [34,35,40,53,54]. This result affirms our previous results, indicating that these extracellular carboxylates play a significant role in defining the electrostatic environment of the extracellular vestibules, constituting a relevant regulatory mechanism of AQPs [34]. Adding more weight to this observation, the “down state” of ar/R arginine and its impact on the water flux of AQP2 was also described by another team through a molecular dynamics study [40]. Even though the authors did not mention the role of ions nature, all their atomic systems were built with 150 mM of NaCl.

Aquaporins were identified in the inner ear many years ago [55], in particular in the endolymphatic sac [17,19,21,56], and are suspected to play a role in the dysregulation of the endolymph barrier resulting in EH. In 1998, AQP2 was already suspected to be involved in MD pathogenesis [56]. Different drugs have been thought to alter AQP function, such as the antidiuretic hormone [15,56,57,58]. AQP2 especially is well known for its role in renal homeostasis under the control of the arginine vasopressin hormone (AVP) [59,60,61], and this regulatory mechanism seems conserved in the inner ear as well. Indeed, levels of AQP2 mRNA and water homeostasis of the endolymphatic sac have been shown to be modulated by application of AVP or vasopressin type 2 receptor antagonist (OPC-31260) [15,62,63,64] highlighting the regulatory role of AQP2 in the inner ear homeostasis. Other AQPs have been suspected to be involved in the pathogenesis of MD, showing a likely key role of this type of membrane channels in the pathophysiology of MD [65,66,67]. Beside the evidence of the role of AQP in endolymph homeostasis, corticosteroids (which have been shown effective in the treatment of MD [24,25,26,27]) have already been suspected to act through the function of AQP, in particular by upregulating mRNA of AQP1 [68], or AQP 3, known to be crucial in the reabsorption of water in the inner ear [69]. Here, we bring strong evidence that dexamethasone can also alter AQP2 function in directly impeding water fluxes through this channel, which has never been shown so far, to our best knowledge. This finding thus draws additional interest to this drug, and may explain part of its efficacy in MD treatment.

The first trials of dexamethasone intratympanic administration for MD were conducted in the 1980s [70] on the premise that the pathophysiology of MD may be immune-mediated [71,72,73]. Initially prescribed with the aim of reducing potential inflammation within the vestibular end organs, hypotheses on the mechanisms of the action of dexamethasone have been enriched over time, with modulating actions of epithelial sodium transport [74], or direct modulation of fluid fluxes [26,28,29,30]. However, none of these hypotheses have been confirmed so far. A direct action of dexamethasone on the control of water fluxes between inner ear liquid compartments is all the more appealing given that endolymphatic hydrops has become a hallmark of this pathology [4,5,6,75,76]. MRI analysis of the inner ear cavities following the so-called “hydrops” protocol (use of contrast enhancers between perilymph and endolymph) now makes it possible to almost systematically correlate increases in the volume of the endolymph, in certain areas of the vestibule, to the symptomology of MD [76]. Therefore, pharmacological approaches to preventing or reducing the water accumulation in the endolymph are expected to be beneficial.

The demonstration that a direct molecular interaction between dexamethasone and AQP2 is structurally possible, and that this interaction can significantly alter the fluxes of water through AQP2, constitutes an important step in understanding the possible mechanism of action of dexamethasone in the treatment of Ménière’s disease. It also opens new avenues in the development of novel pharmacological approaches for a better management of the pathology.

## Figures and Tables

**Figure 1 biomolecules-12-00511-f001:**
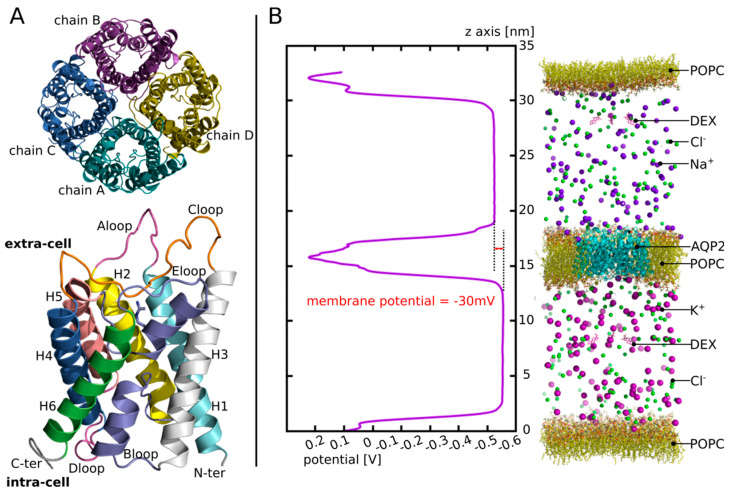
AQP2 structure and illustration of the third experimental setup. (**A**) Schematic representation of the tetrameric assembly of AQP2 (on top) and of the structural features of each monomer (on the bottom) extracted from human AQP2 crystallographic structure (PDB code: 4nef). The tetramer is represented as viewed from the extracellular compartment and the monomer from a lateral view. On the monomer, the B-loop and the E-loop, coloured in purple, meet at the centre of the monomer. They both incorporate a small alpha helix (HB and HE, not mentioned on the legend). One of them holds the arginine of the ar/R constriction (whose side chain is represented) situated on the extracellular half of the channel. (**B**) On the left: the potential along the *z*-axis of the simulation box is displayed. The whole 250 ns of the production phase are taken for membrane potential calculation. On the right: schematic representation of the atomic system simulated. For clarity purposes, water molecules are not represented.

**Figure 2 biomolecules-12-00511-f002:**
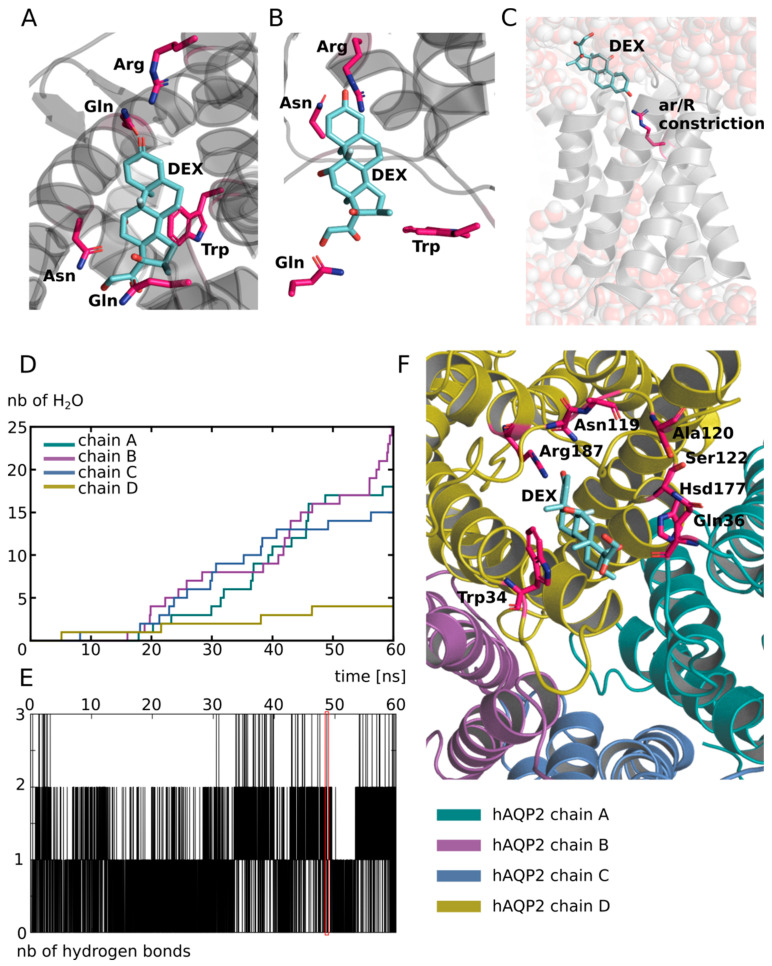
Putative interaction site of dexamethasone with human AQP2. (**A**) Schematic representation of dexamethasone in its native interaction site. The representation is made from the X-ray structure of the agonist form of mouse glucocorticoid receptor obtained from the protein data bank (pdb: 3mne [44]). (**B**) Schematic representation of a dexamethasone molecule positioned manually inside the extracellular vestibule of human AQP2 (pdb: 4nef) in order to mimic its native interaction with its receptor. Four residues are notably identical between the two proteins dexamethasone interaction sites: an arginine, a tryptophane, an asparagine and a glutamine. These residues were used to guide the manual placement of dexamethasone inside the extracellular compartment of AQP2 and are indicated in bold. (**C**) Schematic representation of one protomer of the first experimental setup atomic system at simulation time t = 48.6 ns. The dexamethasone interacts with the AQP through a hydrogen bond with the arginine of the ar/R constriction. Water molecules are represented with spheres. An interruption of the water molecules continuum is visible. (**D**) Cumulative number of water molecules crossing the conducting pore of the channel for each subunit along the 60 ns of simulation. A permeation event is considered when the molecule crosses from end to end a 30 Å long cylinder centred on the centre of the channel which corresponds to the whole membrane cross section. (**E**) Number of hydrogen bonds between the dexamethasone molecule positioned in the extracellular vestibule of chain D and the protein as a function of time. The red box indicates the trajectory frame used as a start point for the other experimental setups and corresponds to simulation time t = 48.6 ns. (**F**) Schematic representation of a dexamethasone molecule in interaction with AQP2 extracted from the simulation (time = 48.6 ns). All of the protein residues involved in the formation of a hydrogen bond along the 60 ns of simulation are coloured in pink. All representations were made with PyMOL software [51].

**Figure 3 biomolecules-12-00511-f003:**
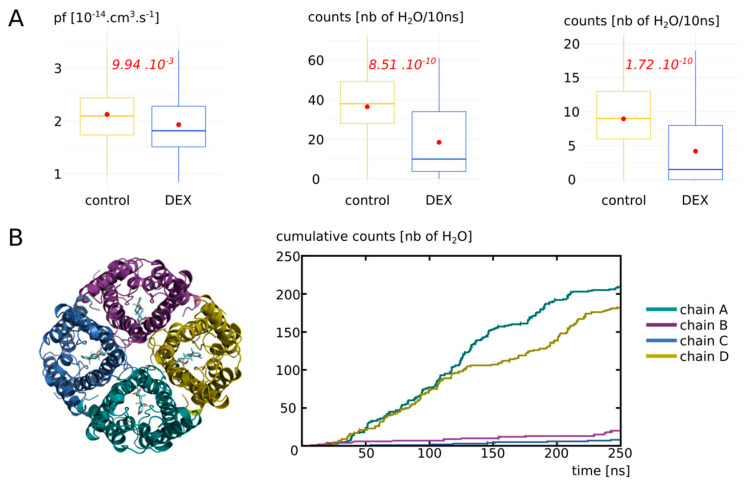
Impact of dexamethasone interaction with AQP2 on the water permeability. (**A**) Water permeability is compared between “control” condition (without dexamethasone) and “DEX” condition (with dexamethasone) through different approaches. From left to right: *pf* derived from the collective diffusion model; number of water molecules crossing a 5 Å section containing the ar/R constriction; number of water molecules crossing the whole conducting pore spanning the entire transmembrane region of 30 Å. Student’s t-test was used to compare *pf* values and Mann–Whitney test to compare counts. *p* values are indicated in red. (**B**) On the left, schematic representation of the tetramer of AQP2 with the four dexamethasone molecules positioned in the extracellular vestibules at the beginning of the simulation in condition “DEX”. On the right, cumulative number of water molecules crossing the whole transmembrane region of 30 Å for each protomer of the condition with dexamethasone (“DEX”) along the whole trajectory.

**Figure 4 biomolecules-12-00511-f004:**
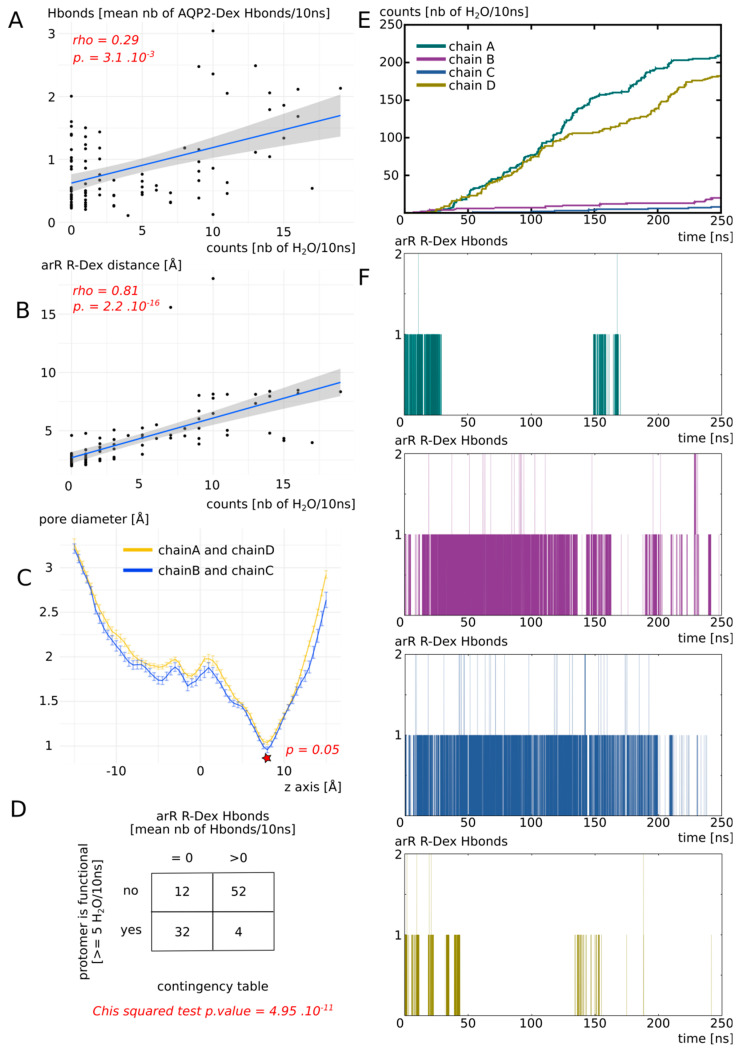
Dexamethasone impacts AQP2 water transport through its interaction with ar/R constriction arginine. All the analysis displayed on this Figure were made from the “DEX” condition of Figure 3. (**A**) Number of water molecules crossing the whole transmembrane section (30 Å long) as a function of the number of H bonds between each dexamethasone molecule and AQP2. Spearman’s rank rho coefficient was used to estimate the correlation between the two variables. (**B**) Number of water molecules crossing the whole transmembrane section as a function of the minimal distance between each dexamethasone molecule and the arginine of the ar/R constriction. Spearman’s rank rho coefficient was used to estimate the correlation between the two variables. (**C**) Mean pore diameter for chain A and D (for which dexamethasone leaved the interaction site) and chain B and C (for which dexamethasone stayed stabilized in the interaction site). The standard error is represented. The star indicates the narrowest part of the channel, the ar/R constriction, where a significant difference was found between the two conditions (Mann-whitney test). (**D**) Contingency table of the functional state of the protomer (defined as functional when allowing 5 or more water molecules to cross the whole transmembrane section in 10 ns), with regard to the presence or absence of H bonds between dexamethasone and the arginine of the ar/R constriction. (**E**) Cumulative number of water molecules crossing the whole transmembrane section as a function of simulation time. (**F**) Number of H bonds between the arginine of the ar/R constriction and dexamethasone for each protomer as a function of simulation time.

**Figure 5 biomolecules-12-00511-f005:**
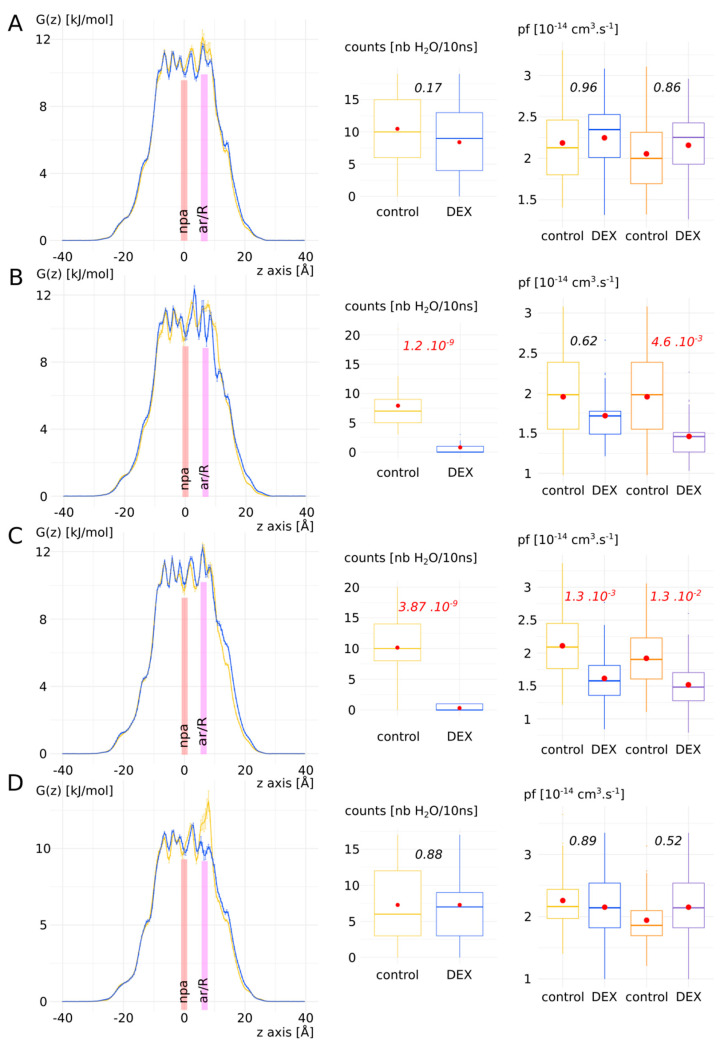
Detailed impact of dexamethasone interaction with AQP2 on the water permeability as an illustration of *pf* adjustment with the *Dk* constant. Chains A–D are displayed from left to right: free energy profiles, number of water molecules crossing the whole transmembrane section (30 Å long) and the *pf* without (colours blue and yellow) and with (colours orange and purple) correction with the *Dk* constant. Control condition without dexamethasone is represented in yellow and “DEX” condition with dexamethasone is represented in blue. For permeabilities comparison between the two conditions, Tukey’s post hoc test after one-way analysis of variance was used for chains A, C and D, and Bonferroni post hoc correction after Wilcoxon test was used for chain B. *p* values are indicated in italic.

**Figure 6 biomolecules-12-00511-f006:**
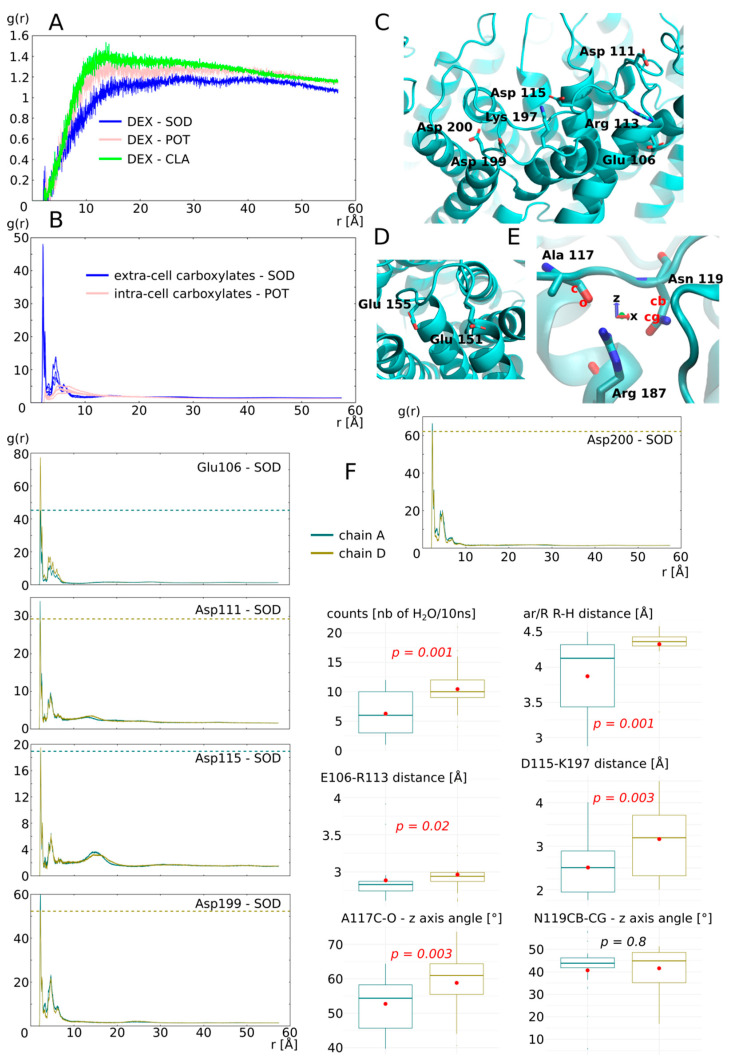
Impact of ion nature on the permeability of the monomers. (**A**) Radial distribution function of ions potassium, sodium and chloride with dexamethasone molecules as a reference calculated from the 5 ns of equilibration phase. (**B**) Radial distribution function of cations with the extracellular surface carboxylates or the intracellular surface carboxylates as reference. (**C**) Schematic representation of the extracellular surface carboxylates plus C-loop arginine113 and E-loop lysine 197 involved in salt bridges with these carboxylates. (**D**) Schematic representation of intracellular carboxylates. (**E**) Schematic representation of the arginine of the ar/R constriction with alanine 117 and asparagine 119 of the C-loop with which it forms hydrogen bonds. The angle formed between Ala 117 atoms C, O and the *z*-axis, and the angle formed between Asn 119 atoms CB, CG and the *z*-axis are used for further characterization of C-loop torsion. (**F**) Characterization of cation nature impact on permeability of the two monomers not interacting with dexamethasone: chain A (in green) and chain D (in yellow). Analyses displayed include: radial distribution function of sodium with each of the extracellular surface carboxylate as a reference; number of water molecules crossing the whole transmembrane section of the channel; minimal distance between the arginine and the histidine of the ar/R constriction; minimum distance between Glu 106 and Arg 113 and between Asp 115 and Lys 197; angle formed between Ala 117 atoms C, O and the *z*-axis and the angle formed between Asn 119 atoms CB, CG and the *z*-axis. All analyses were performed over the whole 250 ns of trajectory. For statistical analysis, chain A and chain D were compared together with non-parametric Mann–Whitney test for averaged values over 10 ns subtrajectories, i.e., 25 repetitions by chain. *p* values are indicated in italic.

**Figure 7 biomolecules-12-00511-f007:**
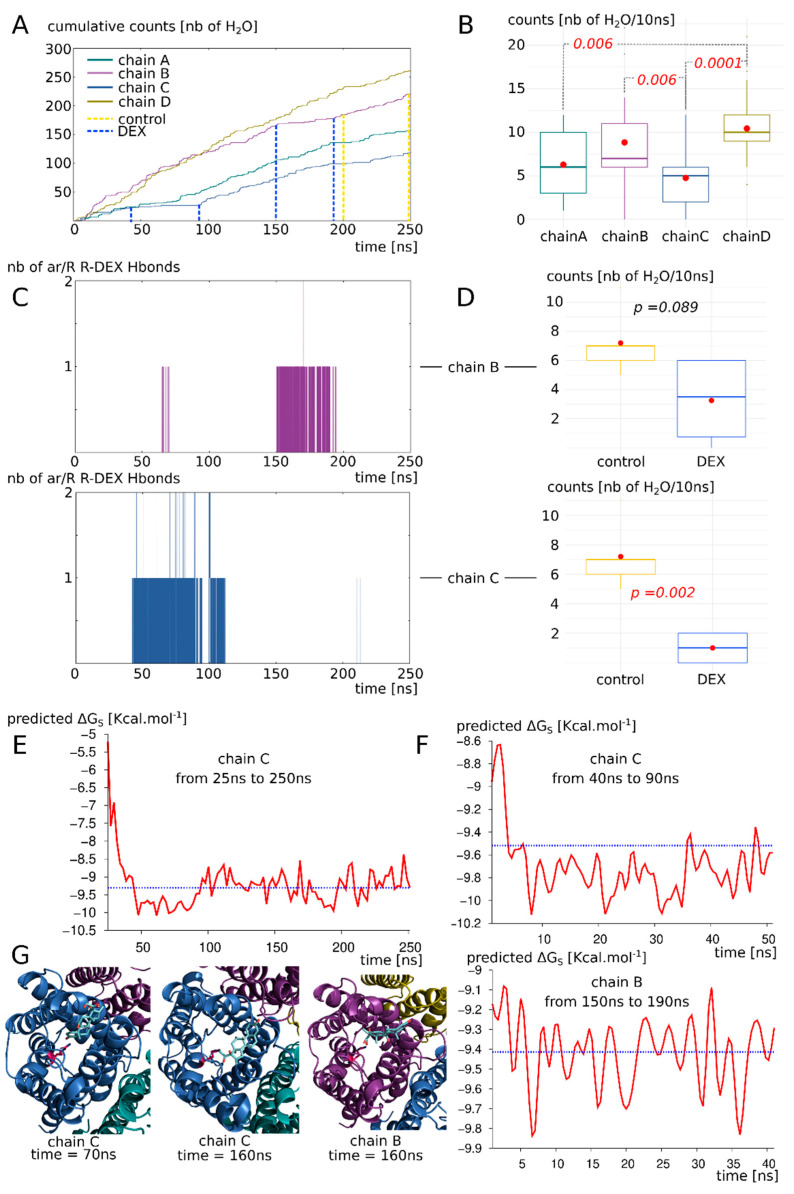
AQP2 interaction with dexamethasone in a biological context mimicking simulation setup. (**A**) Cumulative number of water molecules crossing the whole 30 Å long transmembrane section of each monomer for the complete 250 ns of trajectory. Dashed lines indicate the subparts of trajectory used for statistical analysis of part D and free energy calculation of part F. (**B**) Statistical analysis of the water counts (number of water molecules crossing the 30 Å long transmembrane channel per 10 ns subsection of the trajectory). Non-parametric Wilcoxon test with post hoc Bonferroni correction was used to compare the four chains. Significant differences are indicated by dashed lines with the corresponding *p* value in red italic. (**C**) Number of hydrogen bonds formed between the dexamethasone molecules and the arginine of the ar/R constriction of chain B and chain C. (**D**) Impact of the fixation of dexamethasone on the protomer permeability (chain B on top and chain C on the bottom). The “control” and “DEX” conditions are issued from the same trajectory but at different times based on the presence of absence of interaction with the dexamethasone (indicated by dashed lines on part A). They correspond to 40 or 50 nanoseconds of simulation. T test was performed to compare conditions. *p* values are indicated in italic. (**E**) Binding free energy of the dexamethasone 228 with AQP2—with which it mainly interacts with chain C extracellular vestibule— from time 25 ns to the end of the trajectory at time 250 ns. Mean free energy is indicated by blue dashed line. (**F**) Binding free energy for chain B and chain C subsections (40 ns and 50 ns sections, respectively) corresponding to the parts of the trajectory where hydrogen bond is established between dexamethasone and the arginine of the ar/R constriction. Mean free energy is indicated by blue dashed line. (**G**) Schematic representation of extracellular vestibule of chain B or chain C in interaction with dexamethasone. The arginine of the ar/R constriction is coloured in pink.

**Figure 8 biomolecules-12-00511-f008:**
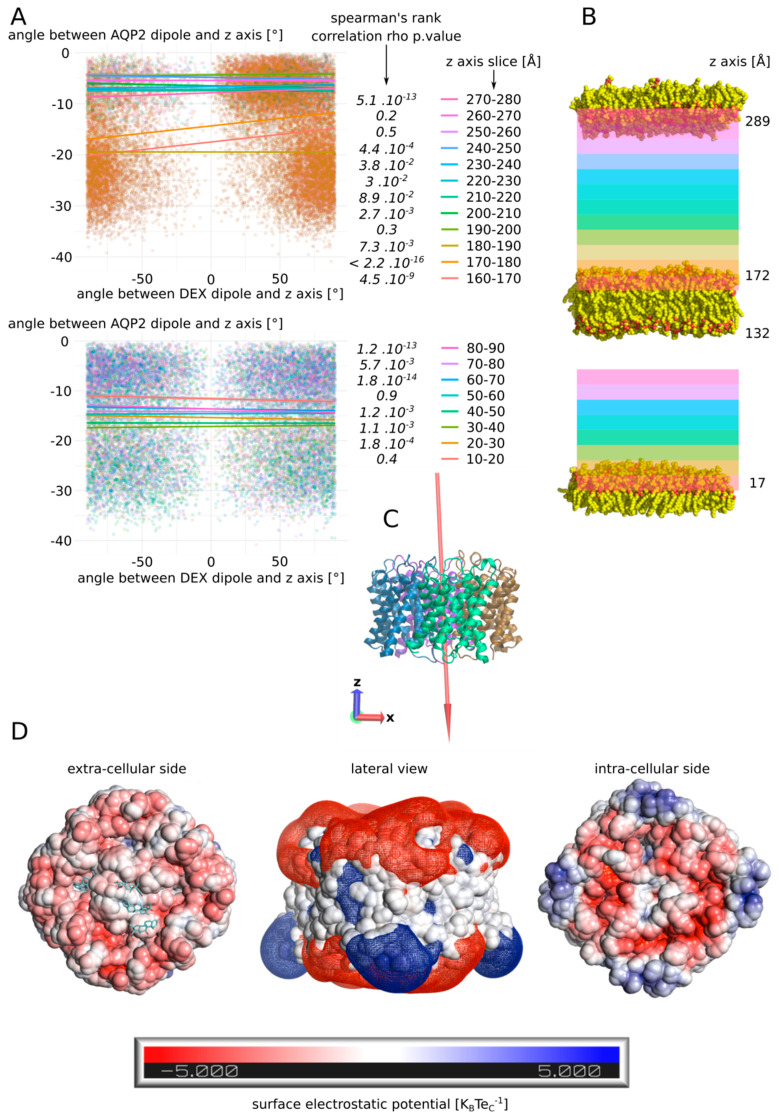
Putative mechanism for spontaneous interaction between AQP2 extracellular surface and dexamethasone. (**A**) Estimation of the correlation between the dipole moment of the whole tetramer of AQP2 and the dipole moment of the dexamethasone molecule. The angle their dipole moment formed with the *z*-axis of the simulation box was calculated. Then, the correlation between angle values for AQP2 tetramer and the dexamethasone was estimated through Spearman’s rank correlation rho. The test was performed according to the position of the dexamethasone molecule in the box. On the upper part, the dexamethasone molecule which interacts with the extracellular vestibule of chain C (dex 228) was followed. On the bottom part, a dexamethasone molecule never interacting with the protein (dex 232) was followed. For this later molecule, data above 90 Å along the *z*-axis were missing so the test could not be performed. *p* values associated with the correlation tests are indicated in italic. (**B**) Schematic representation of the atomic system. The lipids delimiting the two compartments are represented. The z-coordinate of their N or O atoms averaged over the whole 250 ns trajectory is indicated next to the polar heads. The *z*-axis slices used for correlation tests are schematized with colours corresponding to part A. (**C**) Schematic representation of the AQP2 tetramer extracted from the trajectory at time = 250 ns. The dipole moment of the tetramer is schematized by a red arrow (scaling = 0.1). The arrow points from the negative pole toward the positive pole. The *z*-axis and the *x*-axis are represented by red and blue arrows, respectively. (**D**) Surface electrostatic potential of the extracellular and the intracellular faces of AQP2 at time = 50 ns. The dexamethasone molecules interacting with the protein are represented in cyan. In the middle, mesh surface of the electrostatic potential is represented (−0.5 K_B_Te_C_^−1^ in red and +0.5 K_B_Te_C_^−1^ in blue).

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
