# Peer review of "A Perspective for Ménière’s Disease: In Silico Investigations of Dexamethasone as a Direct Modulator of AQP2"

_biomolecules, 2022, doi:10.3390/biom12040511_

Round 1
Reviewer 1 Report
The computational study of Mom et al. represents miscellaneous analysis of potential molecular interactions between dexamethasone, a drug for Meniere’s disease, and aquaporin 2 (AQP2). According to the principal hypothesis underlying this work, dexamethasone is able to decrease the water permeability of AQP2 channel and so macroscopically affect the properties of the fluid compartment of the endolymph leading to the pathology. The authors applied molecular dynamics approaches to propose detailed molecular mechanisms that could be potentially involved in such action of the drug. The methodology used in this work reflects the state-of-art in the field, most of the protocols are clearly and substantially described, proper rigorous statistical analysis supporting the finidngs is conducted, the conclusion are well-grounded. Several different parameters as cumulative number of water molecules passed through the membrane in the simulation, particular hydogen bonds formation, ar/R constriction, free energy profile for the membrane penetration are independently used to demonstrate the same qualitative point: indeed, DEX is able to affect the permeability of the membrane channel and so could play a crucial role in the onset of the Meniere’s disease. Apart from the question directly related to the role of DEX, the impact of ions and the role of specific hydrogen bonds were characterized in a robust way. By all means this laborious high quality study is appropriate to be published in Biomolecules. There are several minor points for the authors to address which could facilitate the reader to better understand the work reported.
MINOR POINTS
1. Quite an important part of the work is related to the modeling of NaCl and KCl salts with the aim to reproduce the conditions more realistically reflecting the studied molecular system. In these lines, mentioning salts in the title would be appropriate.
2. “ar/R constriction” should be defined much earlier in the text, already in the Methods section, where is is mentioned for the first time.
3. In the description of molecular dynamics protocols, it would be useful if the authors mention how many water molecules were used in the simulation, which particular restraints (on which atoms and how strong) were used.
4. p.3, line 113: “Two conditions are yielded this way” should be formulated differently.
5. Section 5 “Conclusions” does not really represents the conclusions drawn from this study but rather the discussion. At the same time, section 4 “Discussion” looks more like a summary/conclusion of the work. At best these sections should be merged and logically connected, otherwise they should be ordered in a different way.
Reviewer 2 Report
My specific comments are
- Title should add molecular dynamics studies to get better idea about the study.
- Abstract need revision particularly the last sentences. It looks like review article abstract instead of research article
- Introduction need more explanation. Author should also introduce little bit about molecular dynamics studies. Author could check recently published article: Bhimaneni, S., & Kumar, A. (2022). Abscisic acid and aloe-emodin against NS2B-NS3A protease of Japanese encephalitis virus. Environmental Science and Pollution Research, 29(6), 8759-8766.
- Every method need proper citation.
- Discussion need revision. Author should discuss key findings with already published studies.
- Conclusion is too long which can be shortened.
- Manuscript should be revise for grammatical and typographical errors
